# Wild Bird-Origin H6N2 Influenza Virus Acquires Enhanced Pathogenicity after Single Passage in Mice

**DOI:** 10.3390/v16030357

**Published:** 2024-02-25

**Authors:** Siqi Tang, Bing Han, Chaofan Su, Hailing Li, Shiyuchen Zhao, Haoyu Leng, Yali Feng, Ying Zhang

**Affiliations:** Key Laboratory of Livestock Infectious Diseases, Ministry of Education, Key Laboratory of Zoonosis, Laboratory of Ruminant Infectious Disease Prevention and Control (East), Ministry of Agriculture and Rural Affairs, Liaoning Panjin Wetland Ecosystem National Observation and Research Station, College of Animal Science and Veterinary Medicine, Shenyang Agricultural University, 120 Dongling Rd., Shenyang 110866, China; 2022200180@stu.syau.edu.cn (S.T.); hbing0430@163.com (B.H.); 2020220609@stu.syau.edu.cn (C.S.); 2018200144@stu.syau.edu.cn (H.L.); zsyc17@126.com (S.Z.); leng1071938690@gmail.com (H.L.); 2020500031@syau.edu.cn (Y.F.)

**Keywords:** wild bird-origin influenza virus, enhanced, adaptation, pathogenicity

## Abstract

The H6 subtype of avian influenza viruses (AIVs) has emerged as one of the predominant subtypes in both wild and domestic avian species. Currently, H6 AIVs have acquired the ability to infect a wide range of mammals, though the related molecular mechanisms have yet to be fully investigated. In this study, a wild bird-origin H6N2 AIV was isolated from the East Asian–Australasian migratory flyway region located in Liaoning Province. This H6N2 virus initially expressed limited replication in mice. However, after one passage in mice, the virus acquired two mutations, PB2 E627K and HA A110V. The mutant displayed enhanced replication both in vitro and in vivo, proving lethal to mice. But the mutant retained the α-2, 3-linked sialic acid binding property and failed to transmit in guinea pigs. We explored the molecular mechanisms underlying the pathogenicity difference between the wild type and the mutant. Our findings revealed that PB2 E627K dramatically enhanced the polymerase activity of the H6N2 virus, while the HA A110V mutation decreased the pH of HA activation. This study demonstrated that the H6N2 subtype wild bird-origin AIV easily acquired the mammalian adaptation. The monitoring and evaluation of H6 wild bird-origin AIV should be strengthened.

## 1. Introduction

Influenza A virus (IAV) has a wide host range, including birds and mammals (including humans), and there is considerable pathogen movement back and forth between reservoir species, posing a significant threat to public health. Wild birds, especially waterfowl, are considered to be the natural reservoir of IAVs and play an important role in spreading influenza viruses worldwide. Certain subtypes of wild bird-origin IAVs such as H3N8 [1], H5N1, H5N6, H6N1 [2], H7N2, H7N3, H7N4, H7N7, H7N9 [3], H9N2 [4], H10N7, and H10N8 [5] have caused human infections [6].

The H6 subtype of avian influenza virus (AIV) is one of the most highly prevalent subtypes among wild birds and poultry. Initially isolated from turkeys in the United States in 1965, it was characterized as a low pathogenic subtype of AIV [7,8]. In China, the H6 subtype AIV was first detected in 1972. Since 2002, H6 AIVs have gradually become one of the prevalent low pathogenic AIVs in live poultry markets, poultry farms, and wild birds [9]. The H6 AIVs have expanded their host range and reassorted with other subtypes during circulation. The H5N8 influenza virus isolated from wild birds reassorted with a local H6N6 domestic duck influenza virus, with the H6N6 virus contributing its PB1 gene [10]. In 2017, two H6N1 AIVs with gene segments originating from H6, H10, H1, and H4 AIVs were found in ducks and wild birds in East Asia [11]. Some of the H6 viruses acquired the ability to bind to human-type receptors, and certain strains even demonstrated efficient transmission ability among mammalian hosts.

There are species barriers between AIVs originating from different hosts. To cross these barriers, certain AIVs can either generate a series of adaptive mutations or reassort with the more adaptable strains to acquire the replication and transmission ability in new hosts. Hemagglutinin (HA) and polymerase basic protein 2 (PB2) are considered the two most important proteins influencing the host adaptability of AIVs [12]. Amino acid mutations in HA, such as Q226L and G228S, enhance the human-type receptor binding efficiency of AIVs. Mutations in PB2, including E627K, D701N, T271A [13], A588V [14], Q591R/K [15], and E158G [14], are associated with increased replication, transmission, and pathogenicity of AIVs in mammalian hosts.

Liaoning Province is located in the East Asian–Australian migratory flyway. During migration seasons, millions of wild birds gather around wetlands in Liaoning, increasing the reassortment and cross-species infection opportunities of AIVs [16]. In this study, we isolated a wild bird-origin H6N2 AIV, A/wild bird/Liaoning/DD535/2021 (DD535), from a wild bird sample during surveillance. This virus expressed limited mice infectivity initially but enhanced infectivity and pathogenicity just after a single passage in mice. A mutant with two mutations, HA A110V and PB2 E627K, was recovered from the infected mice lung. We assessed the biological differences between DD535 and the mutant Mut-DD535 and identified that the mutant acquired higher polymerase activity and lower HA activation pH. These biological characteristics of the mutant were considered to be the determinants of its higher mammalian adaptation and virulence.

## 2. Materials and Methods

### 2.1. Cells and Viruses

Madin Darby canine kidney (MDCK) cells were cultured in Dulbecco’s modified Eagle’s medium (DMEM, GIBCO, Grand Istand, NY, USA) supplemented with 5% fetal bovine serum (FBS, Cellmax, Beijing, China). Human lung carcinoma (A549) and human embryonic kidney 293T (HEK293T) cells were cultured in DMEM supplemented with 10% FBS. All cells were cultured in 5% CO_2_ at 37 °C.

Wild bird fecal samples were tested by RT-PCR based on the influenza virus matrix gene. The HA and neuraminidase (NA) genes of positive samples were amplified and sequenced first to identify the subtype. Subsequently, the supernatants of positive samples were inoculated into 9-day-old embryonated specific-pathogen-free (SPF) chicken eggs. The isolated virus was biologically purified three times by limiting dilution in SPF eggs and stored at −80 °C until it was used. According to this procedure, a wild bird-origin H6N2 AIV, A/wild bird/Liaoning/DD535/2021 (DD535), was isolated and purified.

The infected mouse recovered mutant, Mut-DD535, was purified two times by plaque picking purification, then amplified in MDCK cells.

The entire genomes of DD535 and Mut-DD535 were amplified with primers, as described previously [17], and then sent for sequencing. The viral genome sequences were confirmed by Sanger sequencing before experiments in this study.

### 2.2. Mouse Study

Groups of eight six-week-old and four-week-old female BALB/c mice (Changsheng Biotechnology, Benxi, China) were lightly anesthetized with CO_2_ and intranasally infected with 10^6.0^ 50% egg infective doses (EID_50_) of DD535 or the mutant, Mut-DD535, in volumes of 50 µL. On day three post infection (p.i.), three mice from each group were euthanized, and their nasal turbinates, lungs, brains, spleens, and kidneys were collected for viral titration in eggs by calculating EID_50_. The remaining five mice in each group were monitored for clinical symptoms and weight changes every day. Mice that lost more than 25% of their body weight were humanely euthanized.

### 2.3. Guinea Pig Study

Hartley strain female guinea pigs weighing 300 to 350 g (Changsheng Biotechnology, Benxi, China) were used in these studies. Three guinea pigs were intranasally inoculated with 300 μL of 10^6.0^ 50% tissue culture infective dose (TCID_50_) of DD535 or Mut-DD535 as the inoculated group. Twenty-four hours later, three naïve guinea pigs were placed into the same cage as the contact group. Every 48 h, nasal washes were collected from all guinea pigs until day 14 p.i. Nasal washes were titrated on MDCK cells by measuring TCID_50_. Sera were collected from all the guinea pigs on day 21 p.i. via femoral artery puncture for hemagglutinin inhibition (HI) assay.

### 2.4. Minigenome Assay

To compare the relative polymerase activities of DD535 and Mut-DD535, the polymerase genes of DD535 and Mut-DD535 and the NP gene were cloned into the pCAGGS protein expression vector. The pPolI-NP-Luci plasmid was constructed under the control of human RNA polymerase I promoter as a virus-like RNA encoding the firefly luciferase gene. The pRL-TK plasmid, expressing Renilla luciferase, was used as a control. All constructed plasmids were sequenced to ensure the absence of unwanted mutations. HEK293T cells were co-transfected with polymerase, NP, pPolI-NP-Luci, and pRL-TK plasmids using Lipofectamine 2000 (Invitrogen, Carlsbad, CA, USA). The transfected cells were incubated for 24 h at 37 °C. The cells were then lysed, and the relative luciferase activity was measured by using a dual-luciferase reporter assay kit according to manufacturer’s protocol (Vazyme, Nanjing, China). Luciferase activities were determined with a Spark^®^ multifunctional enzyme labeler (TECAN, Shanghai, China). Data shown are the mean values with standard deviations for the results of three independent experiments.

### 2.5. Receptor Binding Assay

The receptor binding assay was determined by HA assays with 1% chicken red blood cell (cRBC) and sheep red blood cell (sRBC) suspensions. For sialidase treatment, 90 μL of a 10% cRBC suspension was treated with 10 μL of α2,3-sialidase (50 mU/μL) (TaKaRa, Dalian, China) for 10 min at 37 °C. The sample was then washed two times with PBS, centrifuged at 1500 rpm for 5 min each time, adjusted to a final working concentration (1%) with PBS. For the HA assay, viruses were serially diluted twofold with 50 μL of PBS and mixed with 50 μL of a 1% RBC suspension in a 96-well plate. HA titers were read after 30 minutes of the reaction at room temperature [18]. The A/swine/Jiangxi/261/2016(H1N1) (JX261) and A/chicken/Chongqing/SD001/2021(H5N6) (CQ001), which bind to α2,6-cRBCs and α2,3-cRBCs, respectively, were used as controls [10].

### 2.6. Syncytium Formation Assay

MDCK cells were grown in 12-well plates and infected with viruses at an MOI of 3. At 16 h p.i., the cells were treated with 5 μg/mL^−1^ of TPCK trypsin for 15 min and incubated in prewarmed pH-adjusted PBS (pH 4.8–6.2, increasing by increments of 0.1 pH units) for 15 min. Then, the low-pH PBS was replaced with DMEM containing 10% FBS, and the cells were incubated for 3 h at 37 °C for syncytium formation. After 3 h, the cells were fixed with 4% paraformaldehyde and stained with Gimsa (Solarbio, Beijing, China). Images were taken on the Microscopes and Imaging Systems (Leica, Wetzlar, Germany). To quantify syncytium formation, cell nuclei were counted in five randomly chosen fields and the highest pH at which syncytium formation was recorded [19].

### 2.7. Statistical Analysis

Statistical analysis between different groups was performed using a one-way analysis of variance (ANOVA) test via the GraphPad Prism version 8.0 (Graph Pad Software Inc., San Diego, CA, USA). A difference with a value of *p* < 0.05 was considered statistically significant, while *p* < 0.01 was considered highly statistically significant.

## 3. Results

### 3.1. Molecular Characterization of DD535

DD535 possessed the amino acid sequence PQIENR/GLF at the cleavage site in the HA gene, suggesting that DD535 has low pathogenicity. DD535 had a T160A (H3 numbering) mutation in the HA gene that has been identified to enhance the specificity of binding to the human-type receptor and transmission between guinea pigs [20]. Several key molecular markers in DD535 have been identified to play a crucial role in the pathogenicity and transmission of AIVs, such as PB2 G158E, PB2 G309D [21], PB2 T431M [22], PB2 V598T [23], PA K185R [24], PA N383D [25], M1 N30D, M1 A166V [26], and M1 T215A [27], which have been reported as molecular markers of enhanced pathogenicity found in DD535. However, neither the PB2 E627K nor the PB2 D701N mutation, which associate with adaptation of avian viruses to mammals, were found in DD535.

### 3.2. DD535 Could Replicate in Younger Mice

Mice are an ideal model to evaluate AIVs’ mammalian infectivity [28]. Eight six-week-old BALB/c mice were intranasally inoculated with 10^6.0^ EID_50_ of DD535. Three days p.i., the mice organs were collected. As shown in Figure 1A, no virus was detected in any organs of the infected mice. The remaining mice did not show any symptoms during the two-week observation period, and their body weight did not exhibit significant difference compared with the mock mice.

To determine whether DD535 could replicate in younger mice, eight four-week-old BALB/c mice were inoculated with 10^6.0^ EID_50_ according to the same procedure. As shown in Figure 1B, DD535 was detected in only one inoculated mouse lung with a viral titer of 4.2 log_10_ EID_50_/mL. No virus could be detected in the organs of other mice. No significant body weight changes or clinical symptoms were observed in the remaining five mice. These results indicated that DD535 can infect younger mice but express limited mammalian infectivity.

We then investigated whether the wild bird-origin H6N2 AIV DD535 genome could maintain stability during replication in mice. Through plaque purification, we recovered a viral strain from the infected mouse lung. Ten plaques were randomly picked and sequenced. Two mutations, HA A110V (H3 numbering) and PB2 E627K, were detected in all ten purified viral plaques. No mutations occurred in the other six gene segments of mouse lung recovered viral strain. We named this mutant Mut-DD535.

### 3.3. Mut-DD535 Enhanced Replication Ability in Mammalian Cells

To evaluate whether the acquired mutations affect the replication of DD535, we compared the growth characteristics of Mut-DD535 to DD535 in two mammalian cell lines, MDCK and A549 cells. MDCK and A549 cells were inoculated with 10^6.0^ TCID_50_ DD535 or Mut-DD535, respectively, and the supernatant of infected cells was collected every 12 h for viral titration. As shown in Figure 2, both DD535 and Mut-DD535 could replicate efficiently in MDCK and A549 cells. The viral titers of Mut-DD535 were significantly higher at most of the time points than DD535 in both cell lines, about 2.69 to 182.28 times higher than DD535 in MDCK, and 1.13 to 21,673.27 times higher than DD535 in A549. Both viruses replicated better in MDCK than in A549; they reached their peak titers 12 h later but 10 times higher in MDCK than in A549. These results indicated that the Mut-DD535 enhanced the replication ability in mammalian cells.

### 3.4. Mut-DD535 Increased Pathogenicity in Mice

A higher replication ability of AIV in vitro usually correlates with higher pathogenicity in vivo. To evaluate whether Mut-DD535 could also increase pathogenicity in mice, we performed infection experiments in six-week-old and four-week-old mice, respectively, according to the same procedure. Mut-DD535 could replicate in both six-week-old and four-week-old mice. The virus was detected in one mouse nasal turbinate and all mice lungs. As shown in Figure 3A, Mut-DD535 replicated efficiently in six-week-old infected mice lungs, with the average viral titer reaching 7.08 Log_10_TCID_50_/mL. On the other hand, as shown in Figure 3B, the average viral titer in the lungs of four-week-old infected mice was lower, reaching 6.23 Log_10_TCID_50_/mL.

Noticeable IAV-related clinical symptoms that can be used as indicators of pathogenicity, such as depression, anorexia, and ruffled fur, began to be observed on day two p.i. in both age groups of infected mice. Two mice in the six-week-old and one mouse in the four-week-old group were euthanized on day three p.i. due to losing more than 25% of their body weight. On day four p.i., two more mice were euthanized in the six-week-old group. In the four-week-old group, another mouse was euthanized on day seven p.i. At the end of the experiment, the survival rate of Mut-DD535-infected mice in the six-week-old group was 20% and in the four-week-old group it was 60% (Figure 3C). These results suggest that Mut-DD535 has increased pathogenicity in mice compared with DD535.

In order to determine whether the mutations in Mut-DD535 were stably maintained during viral replication in mice, we sequenced the whole genome of lung and nasal turbinate supernatants recovered viruses. The two mutations, HA A110V and PB2 E627K, were stable and no additional mutation appeared in mice recovered Mut-DD535.

### 3.5. Mut-DD535 Did Not Alter Viral Transmissibility Properties

The replication and transmissibility among mammalian hosts is a critical biological characteristic of potential zoonotic AIV. We thus used guinea pigs as an animal model to evaluate the replication and transmissibility of DD535 and Mut-DD535 [29]. As shown in Figure 4, DD535 and Mut-DD535 could replicate in guinea pigs. We detected viral replication in the nasal wash of two inoculated guinea pigs from the DD535 group on day two p.i. and from one guinea pig on day four p.i. In the Mut-DD535 group, viral replication was detected in the nasal wash of all three inoculated guinea pigs on day two and day four p.i. The viral titers from the Mut-DD535 group nasal wash were 17.89 to 19.77 times higher than the DD535-inoculated group. However, neither DD535 nor Mut-535 could transmit between guinea pigs. Sera were collected from all the guinea pigs on day 21 p.i. Seroconversion occurred in all the inoculated guinea pigs but not in the contact guinea pigs. The transmission study indicated that HA A110V and PB2 E627K mutations did not change the transmissibility of DD535 but enhanced DD535 replication efficiency in guinea pigs.

### 3.6. PB2 E627K Mutation Significantly Increased the Polymerase Activity of DD535

The polymerase activity of AIV correlates with viral replication and pathogenicity [30]. The PB2 E627K mutation is considered to be an important mutation for AIV mammalian adaptation. To determine the impact of PB2 E627K on DD535 RNA polymerase activity, the viral small genome polymerase assay was performed by co-transfecting DD535-PB2 (PB2-627E) or Mut-DD535-PB2 (PB2-627K) together with DD535-PB1, PA, and NP-expressing plasmids. As shown in Figure 5A, the PB2-E627K mutation increased the polymerase activity of DD535 by more than 88-fold in HEK293T cells (*p* < 0.001).

### 3.7. HA A110V Reduced the pH of HA Activation

HA binds to the cellular receptors and initiates the viral infection process. The receptor binding property of HA affects the host adaptation of AIV. We used resialylated red blood cells (RBCs) to determine the receptor binding properties of DD535 and Mut-DD535. As shown in Figure 5B, DD535 and Mut-DD535 bound preferentially to the avian-like receptor, α2,3-linked sialic acid, demonstrating that HA A110V mutation did not alter the receptor binding properties of the virus.

After binding to the cell surface receptor, HA mediates AIV entry into the host cells. Membrane fusion is triggered by lower pH, leading to irreversible conformational change in HA. Compared with avian hosts, the pH in the endosome of mammalian hosts is lower [31,32]. Previous studies have confirmed that a lower HA activation pH was generally associated with higher mammalian pathogenicity [33]. Therefore, we investigated the pH of HA activation in DD535 and Mut-DD535 using syncytium formation assays in MDCK cells. We found that the pH of HA activation for DD535 was 5.6, whereas the Mut-DD535 HA activation pH was 5.4 (Figure 5C,D). These results suggested that HA A110V decreased the HA activation pH, which may contribute to the higher replication and pathogenicity of H6N2 viruses in cells and in mice.

## 4. Discussion

Wild birds are generally considered to be the natural hosts of AIVs. Liaoning is located on the East Asian–Australasian flyway route and possesses several migrating stopovers and wintering areas for wild birds. In this study, we isolated and purified an H6N2 subtype wild bird-origin AIV, A/wild bird/Liaoning/DD535/2021 (DD535), in April 2021 from a wetland in Liaoning. Although DD535 exhibited limited replication ability in mice, it acquired two mutations, PB2 E627K and HA A110V, during the first round of infection. These mutations endowed this H6N2 virus with enhanced polymerase activity and decreased the pH of HA activation, ultimately enhancing lethality to mice.

H6 subtype AIVs belong to the low pathogenic AIV and have become one of the most prevalent subtypes worldwide. Previous studies have demonstrated that some of the H6 AIVs isolated from poultry were able to bind α-2,6 SA receptors and transmit among mammalian hosts [7]. The H6 subtype AIVs have already shown the potential to threaten public health.

Genetic recombination and mutations are considered to be the two major mechanisms of AIVs adapting to new hosts. Three of the four human influenza pandemics of the last century were caused by genetic reassortment of influenza viruses. Recent genetic reassortment events were observed in emerging H5N1 [34], H7N9 [35], H10N8 [36], and H5N6 [37]. Most of these reassortants acquired human infectivity. Genetic mutation could occur rapidly once AIV is introduced into other species. Numerous adaptive mutations were detected during avian-origin viruses spilling into mammalian hosts. For example, the H7N9 AIVs in 2013 in China were nonpathogenic for poultry and mice but obtained the PB2 627K or PB2 701N mutation during replication in ferrets, leading to high lethality in mice and ferrets [38]. Our results indicated that the H6N2 subtype wild bird-origin AIV could acquire two adaptive mutations, PB2 E627K and HA A110V, after only one passage in mice. The mutant strain had increased tissue tropism and pathogenicity in mice. Additionally, it exhibited increased replication ability in mammalian cells and enhanced replication efficiency in guinea pigs.

As early as 1993, PB2 E627K was recognized as a determinant of virulence and host range [39]. Subsequent studies revealed multiple functions of PB2 E627K, such as enhancing AIV polymerase activity, replication, and transmissibility in mammals [40,41,42]. In our study, PB2 E627K increased the H6N2 virus polymerase activity in HEK293T cells. HA affects multiple biological characteristics of AIVs, including receptor binding property and membrane fusion, which have been proved critical in determining the host range and pathogenicity of AIVs. Amino acid positions 226 and 228 in the HA protein are associated with receptor binding properties. Both DD535 and Mut-DD535 contained 226Q/228G, suggesting a preference for binding to the avian receptor. The receptor binding assay in our study also confirmed that HA A110V did not affect the receptor binding property of the H6N2 virus. However, our findings indicated that the HA A110V mutation was responsible for the decreased pH required for membrane fusion, which might contribute to the higher replication and pathogenicity of H6N2 viruses in mice. Whether the HA A110V and PB2 E627K mutations acted synergistically or whether host factors were involved during the mutation acquisition still deserve further investigation.

## 5. Conclusions

In summary, we found that the wild bird-origin H6N2 subtype AIV can infect mice without prior adaptation and can rapidly acquire adaptive mutations, HA A110V and PB2 E627K, in mice. A mutant H6N2 virus with these adaptations after a single passage in mice not only exhibited higher replication ability in vitro and in vivo but also enhanced pathogenicity in mice. We further investigated the effects of HA A110V and PB2 E627K and found that HA A110V decreased the pH of the viral membrane fusion, and PB2 E627K increased the viral polymerase activity. Our research indicated that wild bird-origin H6 viruses have zoonotic potential. Continued surveillance and investigation of the H6 influenza viruses circulating in wild birds are needed.

## Figures and Tables

**Figure 1 viruses-16-00357-f001:**
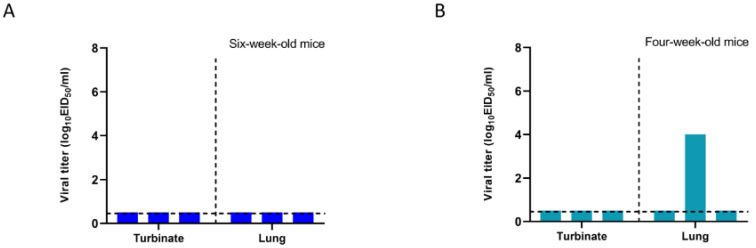
In vivo replication ability of DD535. Three six-week-old mice (**A**) or four-week-old mice (**B**) were lightly anesthetized with CO_2_ and intranasally infected with 10^6.0^ EID_50_ of DD535 in volumes of 50 µL. Three mice from each group were euthanized on day 3 p.i., and their nasal turbinates and lungs were collected and titrated for virus infectivity in eggs.

**Figure 2 viruses-16-00357-f002:**
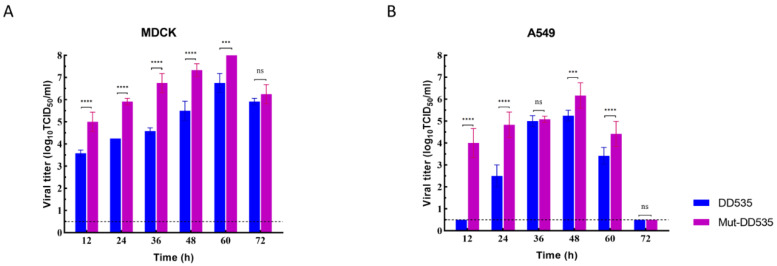
Multicycle replication of DD535 and Mut-DD535 in MDCK and A549 cells. MDCK cells were infected with two viruses at an MOI of 0.001 (**A**), and A549 cells were infected with DD535 and Mut-DD535 at an MOI of 0.1 (**B**). The supernatants were collected at the indicated times and titrated in MDCK cells. The data shown are the means of three replicates, and the error bars indicate standard deviations. Statistical analysis between different groups was performed by using a one-way analysis of variance (ANOVA) test. *** (*p* < 0.001), **** (*p* < 0.0001), and ns (*p* > 0.05).

**Figure 3 viruses-16-00357-f003:**
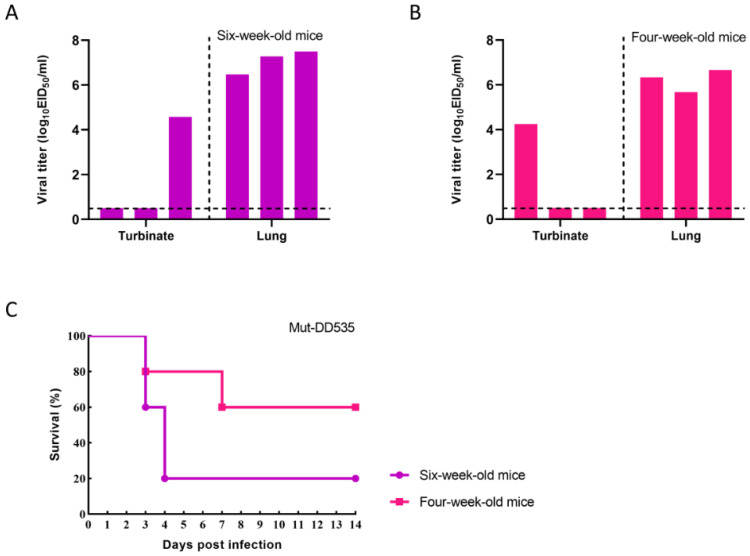
In vivo replication ability of Mut-DD535. Three six-week-old mice (**A**) or four-week-old mice (**B**) were lightly anesthetized with CO_2_ and intranasally infected with 10^6.0^ EID_50_ of DD535-Mut in volumes of 50 µL. Three mice from each group were euthanized on day 3 p.i., and their nasal turbinates and lungs were collected and titrated for virus infectivity in eggs. (**C**) Five six-week-old or four-week-old mice were inoculated intranasally with 10^6.0^ EID_50_ of Mut-DD535 virus. Survival of infected mice was monitored daily for 14 dpi.

**Figure 4 viruses-16-00357-f004:**
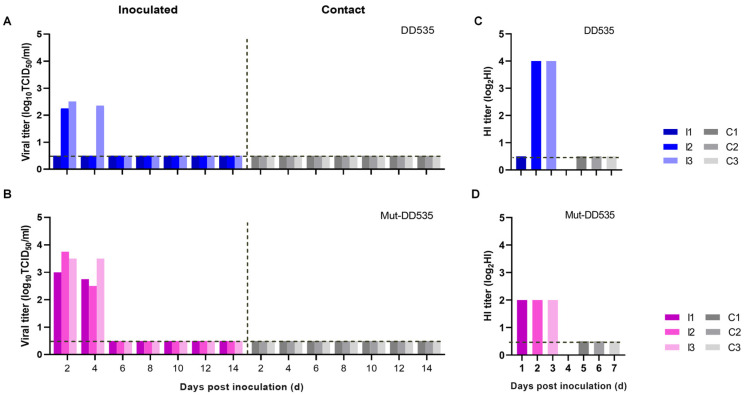
Respiratory droplet transmission of DD535 and Mut-DD535 in guinea pigs. Groups of three guinea pigs were inoculated intranasally with 10^6.0^ TCID_50_ of DD535 (**A**) or Mut-DD535 (**B**). Three naïve guinea pigs were placed in the same cage 24 h after infection as the contact group. Nasal washes were collected from all guinea pigs at the indicated time points for virus detection. Seroconversion of the DD535 (**C**) or Mut-DD535 (**D**) inoculated and exposed animals was confirmed by the use of an HA inhibition test. I—inoculated; C—contact.

**Figure 5 viruses-16-00357-f005:**
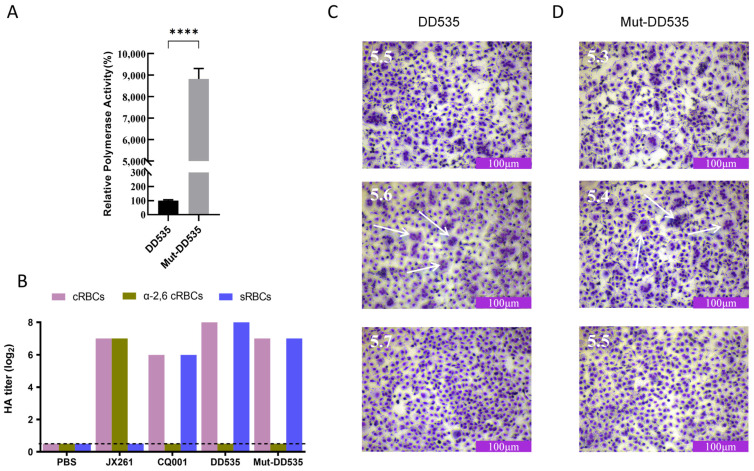
(**A**) The relative luciferase activity of HEK293T cells transfected with pPolI-vRNA, pRL-TK, plasmids expressing DD535-PB2 or Mut-DD535-PB2, DD535-PB1, DD535-PA, and DD535-NP. Statistical analysis between different group was performed by using a one-way analysis of variance (ANOVA) test. **** (*p* < 0.0001). (**B**) Receptor binding analysis of DD535 and Mut-DD535. The three types of red blood cells were cRBCs (1% chicken red blood cells); sRBCs (1% sheep red blood cells); α-2,6-cRBCs (1% chicken red blood cells treated with α2,3-Sialidase). PBS was a negative control group, whereas A/swine/Jiangxi/261/2016(H1N1) (JX261) and A/chicken/Chongqing/SD001/2021(H5N6) (CQ001) were the positive groups binding to α-2,6-cRBCs and sRBCs, respectively. The effect of pH on HA activation of DD535 and Mut-DD535. MDCK cells infected with (**C**) DD535 or (**D**) Mut-DD535 at an MOI of 3 were incubated with pH-adjusted PBS (4.8–6.2). The highest pH at which syncytia formed (arrow) above 50% was defined as the pH threshold.

## Data Availability

The nucleotide sequences of A/wild bird/Liaoning/DD535/2021 (DD535) have been deposited in GenBank under accession numbers: OR905606-OR905613.

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
