# Peer review of "Wild Bird-Origin H6N2 Influenza Virus Acquires Enhanced Pathogenicity after Single Passage in Mice"

_viruses, 2024, doi:10.3390/v16030357_

Round 1

Reviewer 1 Report

Comments and Suggestions for Authors

This paper highlights important work that provided evidence that it may take as little as one passage through a mammal host for an avian influenza to increase its virulence.  Using sound laboratory methods, the authors test the hypothesis regarding drivers for increased virulence of a low-pathogenic avian influenza once in a mammal host. The results of this work are important for understanding all avian influenzas and the risks for adaption to new hosts. There are some minor typos throughout the manuscript that should be addressed in a revision.

Specific Comments

Line 35 …infected humans (remove got) – grammar

Line 38 Should specify here that this is normally a low pathogenic subtype of AIV

Line 44 species barrier – typo

Line 53 Is located in…. grammar past tense

Line 56 Can you describe the isolation technique a little more here

Line 57-62 This is a result of the study and should be moved to the results section

Line 56 instead should be a sentence or two that describes the objectives of this study.  This could also include a description of the specific hypotheses tested between the two groups.  This will help when discussing the statistical analysis later in the manuscript.

Line 94 How was sera collected from the animals?

Line 111 Citations needed

Figure 1 – It might be more sense to have the 4 week old mice be A and the older mice be B

Line 201 Rewrite sentence to make more sense and define ruffle fur

Line 288 Most abundant or common subtypes (typo)

Line 304 – citation needed

Check the grammar throughout the manuscript.

Comments on the Quality of English Language

Check manuscript for minor typos and grammar errors.

Reviewer 2 Report

Comments and Suggestions for Authors

Comments:

This paper describes the comparison of the gene sequence analysis and viral characteristics of the H6N2 subtype of wild bird-derived AIV isolated from wild bird feces, grown in chicken eggs, and tested for infection in mice. In the conclusion of the paper, the authors emphasize the mutation of wild bird-derived AIV of the H6N2 subtype in mice, but they do not deny the possibility that the mutant strain is already present in the original bird feces. 

The comments are added below.

Introduction

1. The wording should be changed because viral mutation and selection of viral strains are included in this phenomenon.

2The possibility that AIV of mix population was selected in mice cannot be ruled out, so the wording of the mouse mutation phenomenon in the conclusion should be changed. Therefore, the present results are insufficient to draw this conclusion.

Material Methods

1, 2.2, Specifically describe the treatment of the samples during genome sequencing.

2, 2.3., Check ethical issues for compliance with animal experiment handling based on the current 3R concept. There is no mention of animal forests. 3, 2.4.

3, 2.4., Inoculation route should be clearly stated.

Result

1, 3.1., Move some quotations to the discussion.

2 It may be understood that the DD535 strain was shown to contain a small number of virus strains with limited infection of mice.

Considerations

1, If the original AIV population had been contaminated with this mouse infectious strain, this result and discussion would change significantly. Therefore, it is difficult to draw conclusions without clarifying the viral diversity of the original fecal AIV.

Reviewer 3 Report

Comments and Suggestions for Authors

Wild bird-origin H6N2 influenza virus acquires enhanced 2 pathogenicity after single passage in mice 3

Siqi Tang, Bing Han, Chaofan Su, Hailing Li, Shiyuchen Zhao, Haoyu Leng, Yali Feng and Ying Zhang

 This is a very nicely done characterization of a wild-bird isolated H6N2 influenza virus.  The virus was not efficient in infecting mice, but was able to infect one 4-week old mouse (out of 8 total infected).  The virus isolated from the lungs of the infected mouse Mut-DD535, showed significantly increased replication in MDCK and 5A569 cells and caused disease in all of the inoculated 4-week and 6-week old mice.  Purified clones of the virus had only two mutations , HA A110V and PB2 E627K.  The authors demonstrated that the HA mutation could be associated with a lowered pH requirement for HA fusion but no change in receptor sialic acid binding specificity.  Moreover the PA mutation was associated with an increased polymerase activity.

The experiments were very nicely done and the controls were all appropriate. The paper is of high interest to the public because of the potential for avian viruses to jump to mammals and the consequent pandemic threat they represent.

I would recommend that the manuscript be published after it is reviewed for English language and shown to reviewers a second time.

Reviewer’s Specific Comments:  A few minor points that should be addressed

Line 31: “Influenza A virus (IAV) has a wide host range therefore posing significant threat to public health.”

·        Please clarify why a “wide host range” poses a significant threat to public health, i.e.,you could mention that the host range includes birds as well as mammals including humans and there is considerable pathogen movement back and forth between reservoir species.

Line 59: Our research indicated that wild bird-origin H6 viruses were on the edge of overcoming the species-barrier, one occasional mammalian host spill over event could endow the virus complete mammalian adaptability.

·        It is not clear what you mean by this statement.  Do you mean one infection of a mammal could lead to a new mammal adapted virus that spreads easily through a mammalian species? I think this statement goes beyond what your paper shows.  Your paper does not demonstrate that one mammalian spillover event could lead to complete adaptation to a mammalian host species. 

Line 121: prewarmed pH-adjusted PBS (pH 4.8–6.2, increasing by increments of 0.1) for 15 min.

Does increasing by increments of 0.1 mean increasing by increments of  0.1 pH units?

Materials and Methods:

·        No method given for viral titration in eggs.

Figure 4: Caption:  it would help to specify that I = infected and C = contact.

Line 272:  This is very confusingly written:

·        It had been reported that the lower HA activated pH related with higher mammalian  pathogenicity.

Discussion

Line 311:  But the HA A110V mutation decreased the pH for HA activation which facilitating the membrane fusion of the wild bird-origin H6N2 virus 312 in mammalian cells.

·        Would be better to say that “the results indicate that HA A110V may be responsible for the decreased pH required for membrane fusion”

Comments on the Quality of English Language

Language problems. 

I cannot recommend manuscript be published as is because it needs considerable editing because of English language usage. This article should be reviewed by an English speaker and the language usage corrected.  Here are a few examples of the extent of the grammatical issues (with my corrections).

Influenza A virus (IAV) has a wide host range and therefore posing poses a significant threat to public health. Wild birds, especially waterfowls are considered to be the natural reservoir of IAV and play important role in spreading influenza viruses world widely. Certain subtypes of wild bird-origin IAVs such as H3N8 [1], H5N1,H5N6, H6N1 [2], H7N2, H7N3, 34 H7N4, H7N7, H7N9 [3], H9N2 [4], H10N7, and H10N8 [5] had got humans infectedhave been responsible for causing human infections [6].

Here is another example:

There are species-barriers between differentAIVs from different hosts origin AIVs. To cross the species - barrier, certain AIVs could either generated a series of mutations or reassorted with other  more adaptable strains to get obtain the ability the to replication replicate and transmission transmit ability in the  new hosts.

Another example:

We used resialylated red blood cells (RBCs) to determine the receptor binding properties of DD535 and Mut-DD535 as described previously [30].

·        The way this is written it sounds as if you are saying that you have determined the receptor binding properties of DD535 and Mut-DD535 previously.   You should write “We used resialylated red blood cells (RBCs) as previously described [30] to determine the receptor binding properties of DD535 and Mut-DD535.

Round 2

Reviewer 2 Report

Comments and Suggestions for Authors

Manuscript ID: viruses-2814796

The revised manuscript generally provides accurate corrections to the comments and makes the content easier to understand. The authors only corrected the areas pointed out in the comments, but missed some relevant points. The research ethics basically requires application approval not only for animal experiments but also for genetic engineering experiments. This should be clearly stated in the section describing experimental methods. The remaining information should be confirmed in accordance with the rules for editing papers.

The description of the virus propagated in the cell line (MDCK, A549), but the relationship to the chicken egg inoculation test is unclear. Also, H6N2 virus is not propagated in MDCK or other cells by this method. It was also explained that the field strain and the mutant strain were passed on by limiting dilution. The method is not described in the method. In the mix population of viruses, when multiple viruses are mixed, the viral load ratio is affected. In other words, please describe the methodology of how much limiting dilution was performed. If the mutation rate of the viruses is the same, it means that certain mutations may occur in mouse passages. That is, if the mouse mutant strain is repeatedly passaged with chicken eggs, will it revert to the bird-derived mutation? I cannot ignore the possibility that the same efficiency of change that occurs in the single passage of mice may cause mutation in successive generations. The basis for proving the change that occurs in one generation of mice is unclear. The issue is to explain and discuss the results of solving this problem.

Although some aspects of the paper are understandable, it is difficult to conclude from the results that the mutation occurred in a single passage of mice, and I would like to comment that the way the paper is presented should be changed.

Reviewer 3 Report

Comments and Suggestions for Authors

Line 18  However, after one generation in mice, the virus acquired two mutations,

·        Should be passage rather than generation.

Authors did not adequately address this comment from my review:

Line 31: “Influenza A virus (IAV) has a wide host range therefore posing significant threat to public health.”

·        Please clarify why a “wide host range” poses a significant threat to public health, i.e.,you could mention that the host range includes birds as well as mammals including humans and there is considerable pathogen movement back and forth between reservoir species.

·        Should be “is one of the most highly prevalent subtypes” or “is one of the prevalent subtypes” depending on whether it is just highly prevalent or just prevalent.

Authors did not adequately address this comment from my review:

Line 40:” The H6 AIVs expanded host range and reassorted with other subtypes during circulation”

·        Give examples of reassortants

Line 66:  Our findings revealed the underlying factors enhancing the pathogenicity of H6 subtype AIVs.

·        I disagree. In my opinion your findings revealed underlying factors enhancing the pathogenicity of this one mutant, not all H6 subtype AIVs. Your findings may suggest potential factors that may affect H6 subtype AIVs in the wild. I would recommend deleting this sentence.

Line 78 Subsequently, the supernatant of positive sample was inoculated into 9-day-old embryonated specific-pathogen-free (SPF) chicken eggs.

·        Should be “the supernatant of positive samples were” or “the supernatant of a positive sample was”

Line 85: The entire genome of DD535 and Mut-DD535 were amplified with primers as described previously [16] then send for sequencing

·        Should be “the entire genomes of were amplified… and then sent for sequencing”

Line 120 (2.5. Receptor binding assay):

·        The authors do not explain how the assay was performed.  The materials and methods should make the results reproducible.

Line 154: “However, PB2 E627K or PB2 D701N mutation were not found in DD535”

·        Should be “However, neither the PB2 E627K nor the PB2 D701N mutation, which have been associated with adaptation of avian viruses to mammals were found in DD535”

Line 169: “We then investigated whether the wild bird-origin H6N2 AIV, DD535, genome could maintain stability during replication in mouse.”

·        Should be “mice”

Line 170: “Through plaque purification, we recovered viral strain from the infected mouse lung.”

·        Should be “recovered a viral strain”

Line 172: “Two mutations, HA A110V (H3 numbering) and PB2 E627K, were detected in all ten purified viral plaques. The mutation rate was 100%.”

·        It is redundant to say that the mutations were detected in all ten plaques and that the mutation rate was 100%.  Delete “the mutation rate was 100%”

Line 190: “they reached their peak titers 12 h latter”

·        Should be “later”

Line 201: “A higher replication ability of AIV usually correlates with higher pathogenicity”

·        A higher replication ability of AIV in cells lines usually correlates with higher pathogenicity in animal models.

Line 202: “To evaluate whether Mut-DD535 also increased mice pathogenicity, we performed mice infection experiments in six-week-old and four-week-old mice”

·        To evaluate whether Mut-DD535 also had increased pathogenicity in mice, we performed infection experiments in six-week-old and four-week-old mice.

Line 217: “These results suggested that Mut-DD535 increased pathogenicity in mice.”

·        “These results suggested that Mut-DD535 had increased pathogenicity in mice compared to DD535.

Line 230: “3.5. Mut-DD535 didn’t change the viral transmissibility”

·        Mut-DD535 does not have altered viral transmissibility properties

Line  231: “The transmissibility among mammalian hosts is a critical biological characteristic of potential zoonotic AIV.”

·        In the Mut-DD535 group, viral replications were detected in the nasal wash of all three inoculated guinea pigs

·         

Line 232: “We then used guinea pigs as an animal model to evaluate the transmissibility of DD535 and Mut-DD535”

·        We thus used guinea pigs as an animal model to evaluate the transmissibility of DD535 and Mut-DD535

Line 236: “In the Mut-DD535 group, viral replications were detected in the nasal wash of all three inoculated guinea pigs

·        “In the Mut-DD535 group, viral replication was detected in the nasal wash of all three inoculated guinea pigs”

In results section 3.5 the term contact group should be used rather than contacted.

Line 254:  “The PB2 E627K mutation is considered to be the prominent mutation for AIV mammalian adaptation”

·        The PB2 E627K mutation is considered to be an important mutation” or the most important mutation if that is what you mean.  Prominent is not the right adjective.

Line 255: “RNP polymerase”

·        Should be RNA polymerase

Line 282: “Therefore, we investigated the pH of DD535 and Mut-DD535 activation using syncytium formation assays in MDCK cells”

·        Should be investigated the pH of HA activation in DD535 and Mut-DD535

Line 294: “Although DD535 exhibited limited mice replication ability in mice”

·        Should be “limited replication ability in mice”

Line 295: “during the first-round infection”

·        first round of infection

Line 301 “The H6 subtype AIVs have already shown potential threaten to public health.

·        “the potential to threaten public health”

Line 307: “Most of these reassortants acquired the human infectivity”

·        “acquired human infectivity”

Line 308: “Numerous adapting mutations

·        “numerous adaptive mutations  

·        Also on line 313

Line 311: “leading to highly  lethality”

·        “leading to high lethality

Line 320: “PB2 E627K conferred the H6N2 virus higher polymerase activity in HEK293T

·        “increased H6N2 polymerase activity”

Line 323: “Amino acid positions 226 and 228 in the HA protein, which are associated with receptor binding properties”

·        Remove “which”

Line 329: “higher mammalian adaptation and pathogenicity”

·        What do you mean by higher mammalian adaption? Increase ability to replicate? Increased titre? Be specific.

Line 330: “any host factors involved”

·        host factors were involved

Line 336: “The mutant”

·        “A mutant H6N2 virus with these adaptations after a single passage in mice ”

Line 339: “Our research indicated that wild bird-origin H6 viruses were on the edge of overcoming the species-barrier and continued surveillance and investigation of the H6 influenza viruses circulating in wild birds was needed.”

·        I disagree.  Your results showed that H6N2 can infect mice after a single passage.  That’s all. It may suggest that there is zoonotic potential, but it is too much to say that H6N2 is on the edge of over coming the species barrier.

·        Should be is needed not was needed.

Comments on the Quality of English Language

Language was much improved but there are still numerous errors, which I corrected in my review.

Round 3

Reviewer 3 Report

Comments and Suggestions for Authors

All comments have been addressed in previous rounds of editing.

Very nice paper!